# Genomic Amplification of *UBQLN4* Is a Prognostic and Treatment Resistance Factor

**DOI:** 10.3390/cells11203311

**Published:** 2022-10-21

**Authors:** Yuta Kobayashi, Matias A. Bustos, Yoshiaki Shoji, Ron D. Jachimowicz, Yosef Shiloh, Dave S. B. Hoon

**Affiliations:** 1Department of Translational Molecular Medicine, Saint John’s Cancer Institute (SJCI), Providence Saint John’s Health Center (SJHC), Santa Monica, CA 90404, USA; 2Max Planck Institute for Biology of Ageing, 50931 Cologne, Germany; 3Department I of Internal Medicine, Center for Integrated Oncology Aachen Bonn Cologne Duesseldorf, Faculty of Medicine and University Hospital Cologne, University of Cologne, 50923 Cologne, Germany; 4Center for Molecular Medicine, University of Cologne, 50931 Cologne, Germany; 5Cologne Excellence Cluster on Cellular Stress Response in Ageing-Associated Diseases, University of Cologne, 50923 Cologne, Germany; 6David and Inez Myers Laboratory for Cancer Genetics, School of Medicine, Tel Aviv University, Tel Aviv 69978, Israel

**Keywords:** Ubiquilin-4, MRE11A, chemotherapy resistance, PARPi, genome amplification, prognosis, DDR

## Abstract

Ubiquilin-4 (*UBQLN4*) is a proteasomal shuttle factor that directly binds to ubiquitylated proteins and delivers its cargo to the 26S proteasome for degradation. We previously showed that upregulated UBQLN4 determines the DNA damage response (DDR) through the degradation of MRE11A. However, the regulatory mechanism at DNA level, transcriptionally and post-transcriptional levels that control *UBQLN4* mRNA levels remains unknown. In this study, we screened 32 solid tumor types and validated our findings by immunohistochemistry analysis. UBQLN4 is upregulated at both mRNA and protein levels and the most significant values were observed in liver, breast, ovarian, lung, and esophageal cancers. Patients with high *UBQLN4* mRNA levels had significantly poor prognoses in 20 of 32 cancer types. DNA amplification was identified as the main mechanism promoting UBQLN4 upregulation in multiple cancers, even in the early phases of tumor development. Using CRISPR screen datasets, *UBQLN4* was identified as a common essential gene for tumor cell viability in 81.1% (860/1,060) of the solid tumor derived cell lines. Ovarian cancer cell lines with high *UBQLN4* mRNA levels were platinum-based chemotherapy resistant, while they were more sensitive to poly (adenosine diphosphate-ribose) polymerase inhibitors (PARPi). Our findings highlight the utilities of *UBQLN4* as a significant pan-cancer theranostic factor and a precision oncology biomarker for DDR-related drug resistance.

## 1. Introduction

Cancer cells have various repairing mechanisms to maintain genomic integrity after DNA lesions caused by exogenous and endogenous factors [1]. DNA double-strand breaks (DSB) are the most toxic lesions in the human genome that can be induced by radiation therapies and chemotherapies [2]. Unrepaired DNA DSBs enhance mutation rate, chromosomal instability, and cell death [3]. Cancer cells have developed essential mechanisms to avoid cell death that enhance the two main DNA DSB repair pathways, homologous recombination (HR) and canonical non-homologous end-joining (NHEJ) [4]. The activation of HR or NHEJ is crucial for repairing DNA damage; however, the balance between HR and NHEJ is critical to determining single nucleotide variant (SNV) rates and genomic stability in cancer cells [5].

Ubiquilins (UBQLNs) family, which includes UBQLN1-4 and UBQLN-L, are key factors in the regulation of protein degradation in cells [6]. All these family members contain a ubiquitin-associated (UBA) domain at the C-terminus and a ubiquitin-like (UBL) domain at the N-terminus. The UBA domain directly binds to ubiquitylated proteins and the UBL domain is responsible for the interaction with the proteasome. UBQLN4 has been identified as a critical molecule functioning as an essential proteasomal shuttle factor that delivers ubiquitylated proteins to the 26S proteasome for degradation [7,8,9]. Previous studies from our group showed that UBQLN4 is activated by the proximal kinase ataxia telangiectasia mutated (ATM). Upon activation, UBQLN4 interacts and controls the protein levels of meiotic recombination 11 homolog A (MRE11A). MRE11A is an essential component of the MRE11–RAD50–NBS1 (MRN) complex and plays a key role in DSB repair, and DNA damage response (DDR) in esophageal squamous cell cancer (ESCC), breast cancer, and neuroblastoma [7,8,9].

As part of the Ubiquitin–proteasome system (UPS), specific proteasome subunits recognize ubiquitylated proteins for degradation [10]. In addition to that, the proteasome shuttle factors provide an additional selectivity by mediating target recognition. UBQLN4 has been reported to be involved in the degradation of a wide range of accessory ubiquitylated proteins [7,8,9,11,12,13,14,15]. Our group has shown that UBQLN4 determines the balance of DDR pathways through the degradation of MRE11A which leads to resistance to DNA damaging agents and sensitivity to poly (ADP-ribose) polymerase inhibitor (PARPi), such as Olaparib in specific cancers [7]. Briefly, our previous studies showed that UBQLN4 binds to ubiquitylated MRE11A, which facilitates ubiquitylated MRE11A mediated 26S proteasome degradation [7,8]. MRE11A degradation repressed HR, while favoring the overactivation of the NHEJ pathway for DNA repair [7]. As a predictive biomarker of drug resistance, *UBQLN4* mRNA levels are associated with the efficacy of neoadjuvant chemotherapy (NAC) including platinum-based chemotherapeutic agents in ESCC [8]. We have shown that elevation of UBQLN4 relates to triple-negative breast cancer (TNBC) and neuroblastoma outcomes [7,9]. However, the regulation of *UBQLN4* overexpression and its utility as a predictive biomarker in various types of solid tumor cancers remains uncharacterized. Therefore, understanding the mechanism at DNA level, transcriptionally and post-transcriptional levels that control *UBQLN4* mRNA levels may provide broad insights and decision-making for precision oncology therapeutic strategies and predictive value for chemotherapy responses inducing DNA damage in specific types of solid cancers.

The aim of this study is to perform a comprehensive pan-cancer analysis of *UBQLN4* using large cohorts of cancer patient datasets. The *UBQLN4* data analysis described in this study has been obtained from 9,962 tumor tissue samples of 32 different solid tumor types and 719 adjacent normal tissues of ten organ types using The Cancer Genome Atlas (TCGA) and 4.918 normal tissues of sixteen organ types using Genotype-Tissue Expression (GTEx) [16]. The relation between *UBQLN4* and treatment resistance was analyzed from 1,060 cell lines of the CCLE database. Our results demonstrated that *UBQLN4* upregulation derived from frequent DNA amplification was observed in various solid tumors. Confirmatory immunohistochemistry (IHC) analyses for UBQLN4 were performed in paired tissue samples from 48 patients. UBQLN4 was associated with poor prognosis across multiple cancer types. *UBQLN4* was detected as a common essential gene across various types of cancer cell lines. High *UBQLN4* mRNA levels have been significantly associated with the DNA repair pathway gene set. Consequently, increased *UBQLN4* levels were associated with cisplatin resistance and Olaparib sensitivity in pan-cancer cell lines.

## 2. Materials and Methods

### 2.1. Analysis of Public Datasets

Datasets from The Cancer Genome Atlas (TCGA) [17] and Genotype-Tissue Expression (GTEx) [16] were downloaded through the University of California Santa Cruz (UCSC) Xena [18]. Datasets from The Cancer Cell Line Encyclopedia (CCLE) [19], Genomics of Drug Sensitivity in Cancer (GDSC) [20], Profiling Relative Inhibition Simultaneously in Mixtures (PRISM) Repurposing Secondary Screen 19Q4 [21], CRISPR gene effect scores inferenced by Chronos [22] and CRISPR gene dependency probability data [23] were downloaded through the Cancer Dependency Map portal (DepMap Portal, http://depmap.org, accessed on 28 April 2022).

### 2.2. Immunohistochemistry

A tissue microarray (TMA, BCN962a) for multiple solid tumor types (AJCC Stage I, II, and III) and respective adjacent normal tissues were obtained from US Biomax (Derwood, MD, USA). The TMA slide was stained with UBQLN4 monoclonal Ab (mAb; #A700-145, Bethyl Laboratories, Montgomery, TX, USA) at the dilution of 1:100 as previously described [9].

### 2.3. DNA Copy Number Analysis

For TCGA analysis, focal-level copy number variation values were generated by using GISTIC2 [24]. For CCLE data analysis, gene-level copy number values utilized were inferred from whole genome sequencing (WGS), whole exome sequencing (WES), or SNP array depending on the availability of the data type. SNP array-based copy number amplification (CNA) profiling was measured experimentally using the Affymetrix Genome-Wide Human SNP Array 6.0 platform at the Broad TCGA genome characterization center and downloaded from the University of California Santa Cruz (UCSC) Xena [18]. In the TCGA analysis, patients with focal *UBQLN4* copy number values larger than 0.3 were defined as *UBQLN4* gene amplified patients.

The CNA analysis in adjacent normal tissues was performed using the TCGA datasets. The data of somatic copy number alterations for 1708 adjacent normal tissues from 27 cancer types in TCGA was obtained from a previous study [25].

### 2.4. Statistical Evaluation of DNA Amplification Recurrence

The recurrence of CNA for each gene was performed using the PART method as previously described [26,27]. In brief, a matrix of CNA profiles based on the TCGA dataset was generated for each cancer type. These matrixes consist of binary values indicating the presence of amplification in each gene and sample, which were indexed in rows and columns, respectively. For each gene, the number of samples with CNA was obtained as a statistical value that measured amplification recurrence and the rate of amplification was calculated for each sample. The significance of the statistical recurrence was examined based on a Poisson binomial distribution using sample-wise amplification rates as parameters [26]. Next, a parameter-based approach was used to calculate the *p*-values for the recurrence of amplification. To calculate *p*-values in the Poisson binomial tests, the Poisson binomial cumulative probability function in the R package poibin was used (http://cran.r-project.org/web/packages/poibin/index.html, accessed on 28 April 2022). Log scaled *p*-values were converted to q-values for multiple testing corrections using the Benjamini-Hochberg procedure [28]. 

### 2.5. CRISPR Screen Datasets

CRISPR screen datasets were obtained from the Cancer Dependency Map portal (DepMap Portal, http://depmap.org, accessed on 28 April 2022) [29]. In brief, gene effect scores were derived from CRISPR knockout screens published by Broad’s Achilles and Sanger’s SCORE projects. A negative score for each gene indicates cell growth inhibition and/or death when its gene is knocked out by the CRISPR screen. Scores are normalized such that nonessential genes have a median score of 0 and independently identified common essentials have a median score of −1, as previously described [30]. Gene Effect scores were inferenced by Chronos [22]. Integration of the Broad and Sanger datasets was performed as previously described [29].

### 2.6. Identification of Common Essential Genes

Common essential genes were identified as previously described [23]. A gene which, in a large, pan-cancer screen, ranks in the top 10 percentile among the most depleting genes in at least 90% of cell lines. The dependency score values were calculated by Chronos and obtained from the DepMap Portal website.

### 2.7. Gene Set Enrichment Analysis

The association between *UBQLN4* mRNA levels from the CCLE dataset [19] and previously defined gene sets were analyzed by gene set enrichment analysis (GSEA). Previously defined gene sets based on their biological functions were obtained from the Molecular Signatures Database v5.2 (http://software.broadinstitute.org/gsea/msigdb/index.jsp, accessed on 28 April 2022). Research manuscripts reporting large datasets that are deposited in a publicly available database should specify where the data have been deposited and provide the relevant accession numbers. If the accession numbers have not yet been obtained at the time of submission, please state that they will be provided during review. They must be provided prior to publication.

### 2.8. Drug Sensitivity Analysis

The area under the dose–response curve (AUC) was downloaded from the DepMap Portal website. AUC values were scaled between 0 and 1 for curves with lower asymptotes less than 1, where lower AUC values indicate increased sensitivity to the treatment.

### 2.9. Identification of Candidate miRNAs That Bind to UBQLN4 mRNA

Four publicly available databases, TargetScan (http://www.targetscan.org/, accessed on 28 February 2022), DIANA TOOL (http://diana.imis.athena-innovation.gr/DianaTools/, accessed on 28 February 2022), miRcode (http://www.mircode.org/, accessed on 28 February 2022), and miRDB (http://mirdb.org/miRDB/, accessed on 28 February 2022) were utilized to determine the miRNAs that bind to the 3′-UTR of *UBQLN4* mRNA.

### 2.10. Biostatistics Analysis

Statistical analyses were performed using R version 4.1.2 (https://www.r-project.org/, accessed on 28 February 2022) in a two-tailed way. The distribution and variation within each group of data were assessed before statistical analysis. Two groups were compared using the Mann–Whitney U test. The correlation was determined using Pearson’s correlation test. In survival analysis using TCGA datasets, patients were divided into high UBQLN4 mRNA expression and low UBQLN4 mRNA expression groups using the minimum *p*-value approach [31]. Overall survival (OS) curves were plotted using the Kaplan- Meier method and *p*-values were calculated using the log-rank test. A *p*-value < 0.05 was considered statistically significant. All figures were unified using Adobe Illustrator CC (Adobe, San Jose, CA, USA).

## 3. Results

### 3.1. UBQLN4 Is Upregulated in Various Types of Cancer

*In silico* analyses were performed using TCGA and GTEx databases to explore how frequent UBQLN4 upregulation was observed in different types of cancers respective to their normal cell origin. *UBQLN4* mRNA levels were upregulated in various types of cancers compared to normal tissues or adjacent normal tissues, which include Liver Hepatocellular Carcinoma (LIHC), Breast Invasive Carcinoma (BRCA), Lung Adenocarcinoma (LUAD), Ovarian Serous Cystadenocarcinoma (OV), Cervical Squamous Cell Carcinoma and Endocervical Adenocarcinoma (CESC), Uterine Corpus Endometrial Carcinoma (UCEC), Lung Squamous Cell Carcinoma (LUSC), Bladder Urothelial Carcinoma (BLCA), Skin Cutaneous Melanoma (SKCM), Esophageal Carcinoma (ESCA), Testicular Germ Cell Cancer (TGCT), Stomach Adenocarcinoma (STAD), Rectal Adenocarcinoma (READ), Colon Adenocarcinoma (COAD), Low Grade Glioma (LGG), Thyroid Carcinoma (THCA), Kidney Renal Papillary Cell Carcinoma (KIRP), and Prostate Adenocarcinoma (PRAD) (Figure 1A,B, and Appendix A). Moreover, *UBQLN1-3* showed upregulation when comparing tumors to normal respective specimens obtained from GTEx, but not when compared to tumor adjacent normal tissue using TCGA datasets (Appendix A). These results were further validated by immunohistochemistry (IHC) in paired analysis in tumor tissue and respective adjacent normal tissues using a tissue microarray (TMA) containing tissue samples from 48 patients. Pan-cancer tumor tissues showed higher UBQLN4 protein levels than paired adjacent normal tissues (Figure 1C). The most significant elevation of UBQLN4 was seen in BRCA, OV, and ESCA (Figure 1C). To summarize, *UBQLN4* mRNA levels are upregulated in 20 of 32 cancer types, resulting in the upregulation of UBQLN4 at protein levels.

### 3.2. UBQLN4 Upregulation Is Associated with Poor Prognosis

Next, we assessed the prognostic significance of *UBQLN4* mRNA levels across different tumor types. Previous studies from our and other groups showed that high *UBQLN4* mRNA levels are associated with poor overall survival (OS) in specific solid tumors [7,8,9,32,33]. Hence, we evaluated the survival rates across various cancer types according to *UBQLN4* mRNA levels. TCGA data analysis demonstrated that patients with high *UBQLN4* mRNA levels had significantly lower OS than patients with low *UBQLN4* mRNA levels in Adrenocortical Carcinoma (ACC), BRCA, COAD, CESC, Kidney Renal Clear Cell Carcinoma (KIRC), KIRP, LIHC, LUAD, Mesothelioma (MESO), Pancreatic Adenocarcinoma (PAAD), Pheochromocytoma and Paraganglioma (PCPG), Sarcoma (SARC), SKCM, and UCEC. (Figure 1D–Q, Appendix A). Of note, patients diagnosed with ACC, KIRC, MESO, SKCM, and UCEC cancer types and advanced—American Joint Committee on Cancer (AJCC) stage III and IV tumors who had high *UBQLN4* mRNA levels showed a worse prognosis than patients with low *UBQLN4* mRNA levels (Appendix A). These results indicate that *UBQLN4* mRNA levels are significantly upregulated and associated with disease outcomes and tumor progression in specific solid tumors.

### 3.3. MiRNA and DNA Methylation Do Not Significantly Control UBQLN4 Gene Expression

Based on the above studies (Figure 1 and Appendix A), we performed a comprehensive analysis of single nucleotide variants (SNV), DNA methylation, and miRNA profiles to investigate potential mechanisms regulating the *UBQLN4* gene. First, we focused on *UBQLN4* SNV. Only 194 functional SNVs of the *UBQLN4* gene in 40,549 patients for 40 cancer types (0.48%) and 86 functional SNVs of *UBQLN4* in 10,953 patients (0.79%) in 32 cancer types were detected using the Catalogue of Somatic Mutations In Cancer (COSMIC) and TCGA datasets, respectively (Appendix A). Thus, *UBQLN4* SNVs are rare and do not explain *UBQLN4* mRNA upregulation.

We then focused on the epigenetic regulation of *UBQLN4.* The TCGA HM450K methylation dataset contains 18 probes allocated across genomic regions of the *UBQLN4* gene (Appendix A, Top). One probe targets a CpG site located in the gene body, another CpG site is located in the 3′ untranslated region (UTR), and the other sixteen probes are designed for a CpG island located in the promoter region of *UBQLN4*. Overall, the two CpG sites located in the gene body and 3′ UTR showed DNA hypermethylation, while the rest of the CpG sites in the promoter region showed DNA hypomethylation (Appendix A, Upper). All the cancer types analyzed did not show statistical differences in DNA methylation profiles between tumor and normal tissue samples. Additionally, the DNA methylation levels at the promoter region of the *UBQLN4* gene were not significantly correlated with *UBQLN4* mRNA levels (Appendix A, Lower). These findings demonstrated that DNA methylation does not have a major role in controlling the *UBQLN4* mRNA levels across different solid tumors.

A previous study in hepatocellular carcinoma (HCC) showed that miR-370 decreases UBQLN4 mRNA/protein levels [32]. Additionally, UBQLN4 overexpression reversed tumor suppressive functions mediated by miR-370 in HCC. We performed in silico analysis and did not identify miR-370 as common miR targeting *UBQLN4* (Appendix A). Expression levels of miR-370 showed statistically significant but only small positive correlation values with *UBQLN4* mRNA levels in the pan-cancer comparisons (Appendix A, Pearson correlation coefficient = 0.18, *p* < 0.001). Focusing on individual cancers, PAAD and READ showed a downregulation in miR-370 and a negative correlation between miR-370 and *UBQLN4* levels, but the correlation values were not significant or statistically significant but small (Appendix A**,** PAAD: Pearson correlation coefficient = −0.28, *p* < 0.0001, READ: Pearson correlation coefficient = −0.0092, *p* = 0.26). In silico analysis also identified miR-7-5p as the only common miRNA targeting *UBQLN4* (Appendix A). MiR-7-5p demonstrated no significant correlation values with *UBQLN4* mRNA levels in pan-cancer comparisons (Appendix A, Pearson correlation coefficient = −0.021). Thus, in silico analysis does not support the possibility that miRs or genome DNA methylation are significant regulators of *UBQLN4* mRNA levels in a pan-cancer analysis.

### 3.4. Copy Number Amplification in The Early Phases of Cancer Induces UBQLN4 Upregulation

*UBQLN4* is localized on chromosome 1q22, a region commonly amplified in solid tumors [8]. Solid tumor tissues from 32 cancer types and adjacent normal tissues from 27 cancer types in the TCGA databases were screened and analyzed to determine the frequency of *UBQLN4* copy number variation (CNV). A high proportion of samples in various types of cancers had CNA for *UBQLN4* (Figure 2A). Other UBQLN family members 1, 2, and 3 were assessed for CNA, even though UBQLN3 is specifically expressed on testes. Lower frequencies for CNA were observed for *UBQLN1-3* (Appendix A). Moreover, the CNV for *UBQLN4* positively correlated with *UBQLN4* mRNA expression in the following solid tumors: UCS, LIHC, CHOL, BRCA, OV, LUAD, CESC, SKCM, LUSC, ESCA, and UCEC (Figure 2B and Appendix A, Pearson correlation coefficients = 0.77, 0.63, 0.83, 0.66, 0.66, 0.69, 0.63, 0.67, 0.60, 0.64, respectively). Furthermore, we estimated the frequency of CNA for each chromosomal position in individual cancers using single nucleotide polymorphism (SNP) array-based CNV profiling. As shown in Figure 2C, the chromosomal region 1q22, which contains *UBQLN4*, was identified as the frequently amplified chromosome region in all these cancer types (Figure 2C, FDR *q*-value < 0.0001). In contrast, the CNVs in adjacent normal tissues were observed in 4.6% (78 of 1,708) of patients (Appendix A). As for the chromosome 1q region, 1.1% (18 of 1,708) of patients had CNVs in 1q. The CNV gains in 1q in the adjacent normal tissues were detected only in nine patients (Appendix A). These findings strongly suggested that DNA amplification controls *UBQLN4* overexpression.

UBQLN4 protein levels positively correlated with the CNV and mRNA levels in the Cancer Cell Line Encyclopedia (CCLE) datasets (Figure 2D). Data integration showed that in patients with LIHC, BRCA, LUAD, UCEC, SKCM, and CESC cancer types, *UBQLN4* gene amplification positively correlated with enhanced *UBQLN4* mRNA levels and is a prognostic factor for OS in all stages (Figure 2E). Collectively, *UBQLN4* upregulation can be explained by genomic DNA amplification and have prognostic utility in different solid tumors.

Interestingly, *UBQLN4* mRNA levels did not change across different histopathology subtypes, such as 1) among the different subtypes of BRCA, and 2) between adenocarcinoma and mucinous carcinoma in COAD/READ and, 3) between esophageal adenocarcinoma (ESAD) and ESCC (Appendix A). Additionally, *UBQLN4* mRNA levels did not significantly change across different AJCC stages for different tumors (Appendix A). Of note, *UBQLN4* amplification and overexpression were observed in individual tumor types from patients diagnosed even with AJCC stage I (Appendix A). These results suggest that *UBQLN4* amplification may occur in the early stages of cancer. For example, patients with LIHC showed the highest *UBQLN4* amplification rate. Furthermore, patients with *UBQLN4* amplification in stage I or stage I/II had a worse prognosis compared to patients with no *UBQLN4* amplification (Appendix A). These results support *UBQLN4* as a potential pan-cancer diagnostic biomarker to assess aggressive early-stage LIHC lesions that will progress or develop resistance.

### 3.5. UBQLN4 Is an Essential Gene for Pan-Cancer Cells

We assessed the importance of UBQLN4 across multiple established solid tumor cancer cell lines using CRISPR screen datasets [22]. In total, 81.1% of the cell lines (860 of 1060) showed high dependency scores on UBQLN4, but not on UBQLN1, UBQLN2, or UBQLN3. Furthermore, UBQLN4 was identified as a common essential gene, and the dependency scores were ranked in the top 10 percentile across all genes in at least 90% of the cell lines (Figure 3A–E), suggesting a key role in cancer cell lines establishment and perpetual survival. We previously demonstrated that ubiquitylated-MRE11A at the DNA damage site interacts with UBQLN4 during DNA damage induced by cisplatin, to be efficiently targeted to the proteasome for degradation, and alternative DDR pathways are consequently activated [8]. MRE11 also showed high dependency scores (Figure 3D,E), but the dependency scores of UBQLN4 and MRE11 were negatively correlated in various solid cancers (Figure 3F,G). This result indicated that MRE11A is not necessarily required for cancer cell lines which highly depend on UBQLN4 expression. Taken together, these results indicate that UBQLN4 plays an essential role in cancer cell lines growth and perpetuation suggesting that genomic alterations promoting UBQLN4 mRNA levels are also conserved in in vitro models.

### 3.6. UBQLN4 mRNA Levels Predict Cisplatin and Olaparib Responses

Our results have demonstrated that *UBQLN4* upregulation promotes NHEJ by increasing MRE11A degradation [7,8]. Thus, *UBQLN4* levels may have potential utility as a predictive biomarker for the usage of DNA damage drugs. Our previous study showed that UBQLN4 protein levels were significantly higher in tissue biopsies obtained from non-responder than in responder ESCC patients to cisplatin-based neoadjuvant chemotherapy (NAC) [8]. To explore the potential utility of *UBQLN4* as a predictive marker for drug sensitivity in pan-cancer, we examined the association between *UBQLN4* mRNA expression and drug sensitivity using the combined datasets of CCLE and Profiling Relative Inhibition Simultaneously in Mixtures (PRISM) Repurposing Secondary Screen datasets [21]. Cell lines with high *UBQLN4* mRNA levels had significantly altered DNA repair pathways (Figure 4A). Then, we evaluated the sensitivity of ovarian cancer cell lines to cisplatin. Sensitivity was assessed based on area under the curve (AUC) values, meaning that cell lines with higher AUC will be more resistant to treatment. The results showed that cancer cell lines with high *UBQLN4* mRNA levels had higher cisplatin AUC values in all cancer types (Figure 4B). In ovarian cancer cell lines, *UBQLN4* mRNA levels positively correlated with cisplatin AUC values (Figure 4C, Pearson correlation coefficient = 0.58, *p* = 0.0050). Ovarian cancer cell lines with high *UBQLN4* mRNA levels had the lowest sensitivity to cisplatin treatment (Figure 4D, Mann–Whitney U test: *p* = 0.0041).

Previous reports showed that *UBQLN4* upregulation improves the sensitivity to PARPi by repressing HR activity through the degradation of MRE11A [7,8]. Cancer cell lines with high *UBQLN4* mRNA levels were associated with a better Olaparib response (Figure 4E). In ovarian cancer cell lines, *UBQLN4* mRNA levels negatively correlated with Olaparib AUC values (Figure 4F, Pearson correlation coefficient = −0.31, *p* = 0.18). The Genomics of Drug Sensitivity in Cancer 2 (GDSC2) datasets also showed a negative correlation between *UBQLN4* mRNA levels and Olaparib AUC values (Figure 4G, Pearson correlation coefficient = −0.5, *p* = 0.011). Similar analyses were performed only in *BRCA1/2* wild-type ovarian cancer cell lines. No significant differences in *UBQLN4* mRNA levels were observed in BRCA1/2 mutants versus wild-type ovarian cancer cell lines (Appendix A). Additionally, *BRCA1/2* wild-type cell lines with *UBQLN4* high mRNA levels showed significantly lower sensitivity to cisplatin and significantly higher sensitivity to Olaparib (Appendix A). Interestingly, the ovarian cancer cell line OVK18, which showed the highest *UBQLN4* mRNA levels in CCLE had the highest sensitivity to Olaparib, despite having the lowest sensitivity to cisplatin treatment (Figure 4C,F). These results support that *UBQLN4* is a key factor to maintain an important balance of DDR and thus, elevated *UBQLN4* mRNA levels determined resistance to cisplatin, while improving PARPi sensitivity in cancer cell lines.

## 4. Discussion

In this study, we showed that both *UBQLN4* mRNA and protein levels are upregulated in various solid cancers driven by *UBQLN4* gene DNA amplification. Importantly, we demonstrated that *UBQLN4* DNA amplification is a positively selected mechanism in cancer progression and this mechanism is observed frequently even in the early phases of cancer development. A previous study showed that DNA amplification is an early event in cancer with clonal patterns that are preserved in the matched metastatic samples [34]. We previously demonstrated that in ESCC FFPE tissues from endoscopic core biopsies with high *UBQLN4* mRNA expression could predict the worse response to NAC [8]. These results suggested that *UBQLN4* DNA amplification mechanisms in the early stage are acquired clonally within the tumor through cancer progression, thus even small tissue biopsies can be used to detect *UBQLN4* mRNA upregulation. Though tumor heterogeneity remains a major cause of the prognostic and therapeutic difficulty in the management of advanced stages of cancer [35], clonal transcriptomic biomarkers such as *UBQLN4* upregulation can be very useful in precision oncology treatment decisions. This is also important in assessing tumor tissue by biopsies for potential drug resistance. The prognostic utility of *UBQLN4* may represent a potential novel therapeutic target to overcome specific drug treatment resistance.

We demonstrated that cell lines with high *UBQLN4* mRNA levels have significantly altered the DNA repair pathway, and the cell lines that depend on UBQLN4 do not depend on MRE11 for survival. These results are consistent with our previous functional studies indicating that UBQLN4 accelerates the NHEJ pathway through MRE11A degradation. Moreover, the analysis of CRISPR screen datasets showed that UBQLN4 is essential in 81.1% of all cancer cells (860 out of 1060). *UBQLN4* has been reported to have an impact on the degradation of a wide range of ubiquitylated proteins [12,13,36,37,38,39,40]. For example, UBQLN4 targets misassembled ER-localized proteins to the 26S proteasome and harbors a protective role by reducing proteotoxic cell stress [11]. Furthermore, our recent study has shown that *UBQLN4* targets STING in triple-negative breast cancer [9]. Nuclear DNA damage leads to the accumulation of cytosolic-DNA, which in turn leads to activation of the STING pathway, and the following transcription of pro-inflammatory cytokines [41]. Notably, we have shown that UBQLN4 promotes STING degradation during DNA damage—and STING agonist-induced STING activation, and this may be one of the central mechanisms controlling STING downstream activation during treatment [9]. These studies also indicated a potential integration of DNA damage responses, UBQLN4, and immune responses in solid tumors. These previous reports support that UBQLN4 is closely related to DNA repair pathways through the interactions with specific response pathways such as key immune factors. The studies also demonstrate the UBQLN4 has multiple functions in responding to DNA damages [7,8,11,15,42] and regulating protein clearance to prevent proteotoxic cell stress an event that would be quite active in proliferating cancer cells.

Our analysis showed that *UBQLN4* mRNA levels are associated with current chemotherapies resistance in cancer cells such as cisplatin resistance and Olaparib sensitivity in many different cancer cell lines. Interestingly, ovarian cancer cell lines which have high mRNA levels of *UBQLN4* showed cisplatin resistance and Olaparib sensitivity at the same time. Both drugs are used in treating advanced stages of ovarian cancer [43,44]. The same results were observed in *BRCA1/2* wild-type ovarian cancer cell lines. Our findings suggest that *UBQLN4* mRNA levels allowed for triaging ovarian cancer patients that will receive platinum drugs or PARPi therapies. Additional studies are required to (1) validate our findings and determine whether *UBQLN4* mRNA levels can improve clinical decisions in precision medicine therapeutic strategies; and (2) to determine whether the effect of HRD induced by the *UBQLN4* upregulation is synergistic to HRD induced by *BRCA1*/*2* mutations.

## 5. Conclusions

In conclusion, this study provides new evidence that *UBQLN4* mRNA and protein levels are regulated by CNA. UBQLN4 has clinical translational implications as a diagnostic, prognostic, and treatment resistance biomarker for adjuvant platinum-based drugs or PARPi therapies.

## Figures and Tables

**Figure 1 cells-11-03311-f001:**
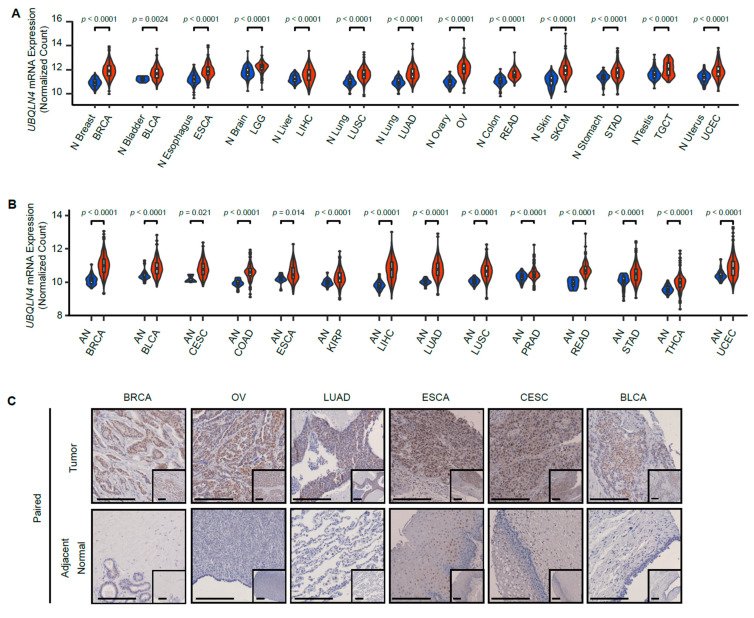
*UBQLN4* mRNA levels are upregulated in various types of solid tumors. (**A**) *UBQLN4* mRNA levels in tumor tissues (TCGA dataset) and normal tissues (GTEx dataset). Statistical differences were calculated by the Mann–Whitney U test. (**B**) *UBQLN4* mRNA levels in tumor and adjacent normal tissues from TCGA dataset. Statistical differences were calculated by the Mann–Whitney U test. (**C**) UBQLN4 protein levels in paired tumor and adjacent normal tissues from BRCA, OV, LUAD, ESCA, CESC, BLCA cancers. IHC analysis in the TMA were performed with UBQLN4 mAb (1:100). The insets represent the whole image of the core in the TMA. Scale bars = 200 µm. The magnifications represent an enlarge image of a specific site in the cores of the TMA. Scale bars = 200 µm. (**D**–**Q**). Kaplan–Meier curves for ACC, BRCA, CESC, COAD, KIRC, KIRP, LIHC, LUAD, MESO, PAAD, PCPG, SARC, SKCM, and UCEC patients according to *UBQLN4* mRNA expression in TCGA datasets. Statistical differences were evaluated using the log-rank test. Abbreviation lists can be found in the materials and methods section.

**Figure 2 cells-11-03311-f002:**
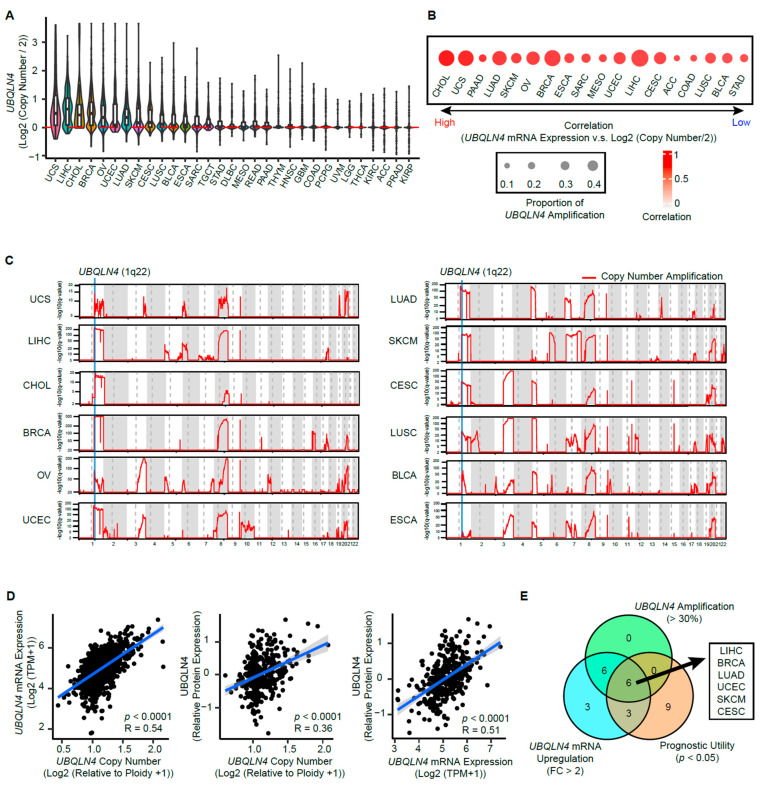
*UBQLN4* showed genomic amplification in various types of solid tumors. (**A**) The distribution of *UBQLN4* DNA copy number in different types of cancer in TCGA. (**B**) The correlation between *UBQLN4* mRNA expression, DNA copy number, and the proportion of *UBQLN4* amplified samples in TCGA. Samples with the values of *log2 (copy number/2)* larger than 0.3 are categorized as amplified samples. Sizes of dots represent the proportion of amplified samples, while colors of dots represent the Pearson correlation coefficient between mRNA expression and DNA CNA of *UBQLN4*. (**C**) Chromosomal regions are significantly affected by copy number amplifications (CNA). Statistical significance for recurrence of CNAs was evaluated by PART analysis. Blue lines indicate the chromosomal region of *UBQLN4* in the 1q22 chromosomal region. (**D**) Pearson’s correlation between DNA copy number, mRNA expression, and protein expression of *UBQLN4* of pan-cancer cell lines in the CCLE datasets. (**E**) Integration of *UBQLN4* analysis. Each number indicates the number of cancer types satisfying each criterion.

**Figure 3 cells-11-03311-f003:**
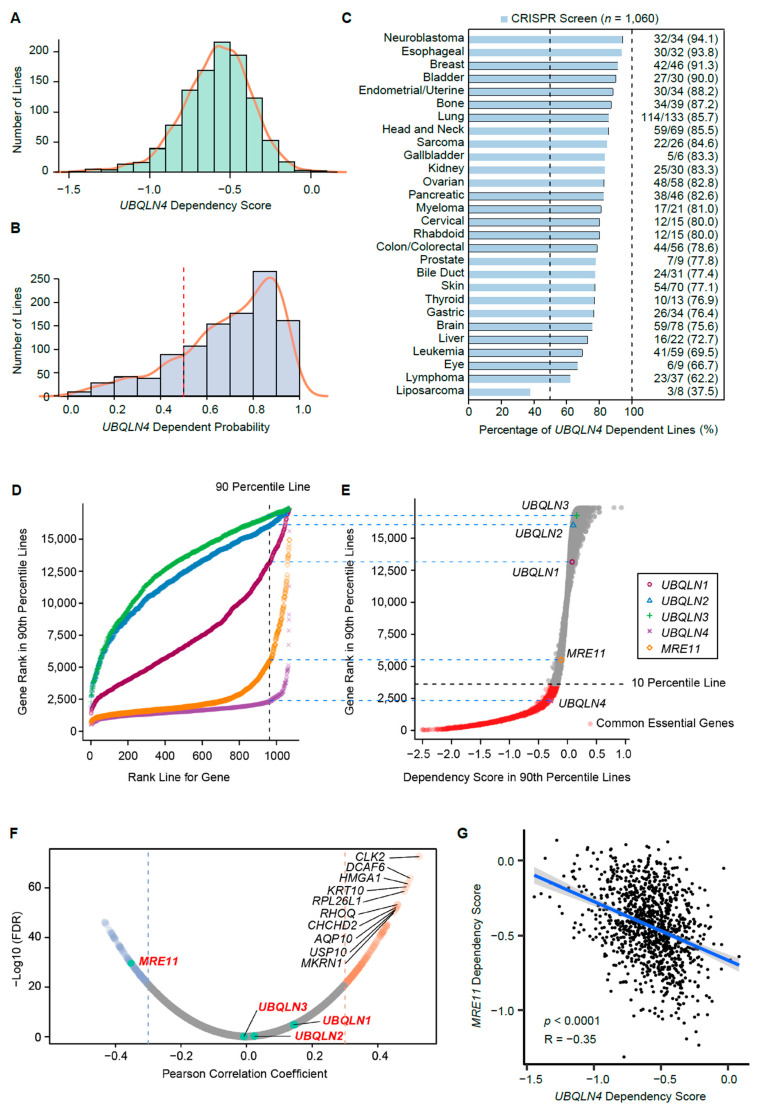
Utility of *UBQLN4* as a biomarker of prognosis and treatment resistance. (**A**) Histogram of *UBQLN4* dependency scores of all cell lines in the Cancer Cell Line Encyclopedia (CCLE) datasets. (**B**) Histogram of *UBQLN4* dependent probability scores of all cell lines in CCLE datasets. (**C**) Frequency of *UBQLN4*-dependent cancer cell lines by different cancer types in the CCLE dataset. Cell lines that have a larger *UBQLN4* dependent score than 0.5 are defined as *UBQLN4* dependent. (**D**) The relationship of *UBQLN1-4* and *MRE11* between a gene’s score rank in a cell line and the cell line’s rank for that gene using dependency scores of all cell lines in CCLE datasets, with gene ranks in their 90th percentile of least dependent lines highlighted. (**E**) Distribution of gene ranks for the 90th percentile of least dependent cell lines for each gene and dependency score in 90th percentile lines. Black dotted lines indicate the top 10 percentile most depleting genes in at least 90% of cell lines. The y-axis is equivalent to the y-axis in D at the 90th percentile mark, as indicated by the blue dotted lines. (**F**) Volcano plot of Pearson’s correlation coefficients between *UBQLN4* gene dependency scores and dependency scores of each gene of all cancer cell lines in CCLE datasets. (**G**) Pearson’s correlation of *UBQLN4* and *MRE11* dependency scores of all cancer cell lines in CCLE datasets.

**Figure 4 cells-11-03311-f004:**
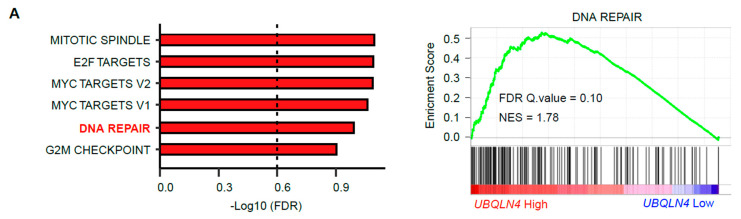
High *UBQLN4* mRNA levels are associated with cisplatin and Olaparib responses. (**A**) All cell lines from CCLE dataset were divided according to the median *UBQLN4* mRNA levels and gene set enrichment analysis (GSEA) was performed to compare the difference between the two groups. (Right) Bar plot showing the six significantly enriched gene sets in high *UBQLN4* mRNA group. The x-axis indicated the minus logarithm of FDR values of each gene set. (Left) Comparison of the normalized enrichment score for the DNA repair pathway determined by GSEA. FDR, false discovery rate; NES, normalized enrichment score. (**B**) Comparison of cisplatin area under curve (AUC) values in low *UBQLN4* expression cell lines and high *UBQLN4* expression cell lines in all cancer types in PRISM Repurposing Secondary Screen datasets. Statistical difference was evaluated using the Mann–Whitney U test. (**C**) Pearson’s Correlation between UBQLN4 mRNA levels and cisplatin AUC values in ovarian cancer (OV) cell lines in PRISM Repurposing Secondary Screen datasets. The red dots indicated the cell lines containing *BRCA1* or/and *BRCA2* mutations. (**D**) Comparison of cisplatin AUC values in ovarian cancer (OV) cell lines with low and high *UBQLN4* mRNA levels in PRISM Repurposing Secondary Screen datasets. The red dots indicated the cell lines containing *BRCA1* or/and *BRCA2* mutations. Statistical difference was evaluated using the Mann–Whitney U test. (**E**) Comparison of Olaparib AUC values in low and high *UBQLN4* mRNA in all cancer types in PRISM Repurposing Secondary Screen datasets. Statistical difference was evaluated using the Mann–Whitney U test. (**F**) Pearson’s correlation between UBQLN4 mRNA expression and Olaparib AUC values in OV cell lines in PRISM Repurposing Secondary Screen datasets. The red dots indicated the cell lines containing *BRCA1* or/and *BRCA2* mutations. (**G**) Comparison of Olaparib AUC values in low and high *UBQLN4* mRNA levels cell lines in OV in Sanger GDSC2 datasets. The red dots indicated the cell lines containing *BRCA1* or/and *BRCA2* mutations. Statistical difference was evaluated using the Mann–Whitney U test.

## Data Availability

All datasets utilized in this study are publicly available.

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
