# Peer review of "Genomic Amplification of UBQLN4 Is a Prognostic and Treatment Resistance Factor"

_cells, 2022, doi:10.3390/cells11203311_

Round 1
Reviewer 1 Report
Yuta Kobayashi and colleagues performed in silico analyses using TCGA and GTEx databases revealing that UBQLN4 is upregulated in multiple types of cancer. Further analyses revealed that UBQLN4 mRNA levels are correlated with upregulation at the protein level, and associated with disease outcomes and tumor progression in multiple solid tumors. In search of the mechanisms for UBQLN4 upregulation they discovered that DNA copy number is closely correlated with expression in multiple cancer types, and appears at early phases of cancer progression. Using a CRISPR screen database they reveal that most cancer cell line are dependent on continued expression of UBQLN4. Finally, they report that ovarian cancer cell lines with high UBQLN4 mRNA levels are resistant to platinum-based chemotherapy while responsive to PARP inhibitors.
This is a well written paper, that nicely extends their previous landmark report in Cell which focused on esophageal SCC (Jachimowicz RD et al Cell 2018) extending the role of UBQLN4 as an important pan cancer biomarker. Furthermore, it provides a potential mechanism for UBQLN4 overexpression in cancer.
Comments:
Increased UBQLN4 levels were associated with cisplatin resistance and Olaparib sensitivity in ovarian cancer cell lines. This is very interesting, as in general, BRCA1 and 2 mutations are known to confer sensitivity to both of these treatments. In fact, in clinical practice it is known that response to platinum agents is a predictor of response to olapraib (see doi: 10.1200/JCO.2009.26.9589). I am not aware of other biomarkers, that similar to UBQLN4 expression can discern between response to these two drugs. It would be interesting to repeat the analysis of the cell lines shown in fig.4, separating the cell-lines into those with and without homologous recombination deficiency.
Minor comments
In fig. 1C the normal tissue in ovary, esophagus and cervix should show more of the epithelial component, which is the equivalent of the epithelial cancers shown. In CESC tumor it looks like an in situ lesion – if they have a picture of a clearly invasive tumor I suggest to replace this. The insets are not explained in the legend.
Lines 252 and 269 – I suggest to tone down the conclusion. Something like – our analysis does not support the possibility that…
I identified some typos (L stands for line):
L48 a ubiquitin (however 'an ubiquitin' is also accepted yet less commonly used, so up to the authors. This appears many times in the text).
L60 – the authors use ubiquitylated, in other parts they use ubiquitinated (e.g. L67). Both are OK, they should choose one of those and be consistent.
L198 from 48 patients
L278 the abbreviation UCS is not explained anywhere.
L383 should be: highest sensitivity to cisplatin treatment
L413 sentence not clear to me
Author Response
Reviewer 1
Comments:
- It would be interesting to repeat the analysis of the cell lines shown in fig.4, separating the cell-lines into those with and without homologous recombination deficiency.
- We thank the reviewer for the comments. As suggested, we separated the cell lines into those with and without homologous recombination deficiency (HRD). There are different gene signatures that have been associated with HRD; thus, making the classification of cell lines based on HRD more complex. The definition of HRD+ cell lines we utilized was based on BRCA1/2 deleterious mutations. No significant differences in UBQLN4 mRNA levels were observed in ovarian cancer cell lines with HRD+ versus HRD-. BRCA1/2 wild type (HRD-) ovarian cancer cell lines showed comparable results to those observed in all ovarian cancer cell lines (HRD+/-). The BRCA wild-type ovarian cancer cell lines with UBQLN4 high mRNA levels showed significantly lower sensitivity to cisplatin and significantly higher sensitivity to Olaparib. Because of the small number of ovarian cancer cell lines from both PRISM Repurposing Secondary Screen datasets and GDSC2 datasets with BRCA1/2 deleterious mutations, we could not examine the association between UBQLN4 mRNA levels and the sensitivity to both cisplatin and Olaparib. Further detailed studies are required to clarify whether the effect of HRD induced by the upregulated UBQLN4 is synergistic to the HRD enforced by BRCA1/2 deleterious mutations in assessment of multiple ovarian cancer tissues and cell lines.
- We have added the new data in Figure S11 and added the information of BRCA1/2 status in Figure 4C, 4D, 4F, and 4G, as well as further explanations of the results in lines 389-394, 468-469, 473-475.
Minor comments
- In fig. 1C the normal tissue in ovary, esophagus and cervix should show more of the epithelial component, which is the equivalent of the epithelial cancers shown. In CESC tumor it looks like an in-situ lesion – if they have a picture of a clearly invasive tumor I suggest to replace this. The insets are not explained in the legend.
- We have replaced the images to address the reviewers’ specific concerns. The images were focused on the epithelial component of the ovary, esophagus, and cervix normal tissues. The in-situ CESC lesion was replaced for a CESC tumor with an invasive component. Please refer to Figure 1C. We added a better explanation for the insets in the legend of Figure 1C.
- Lines 252 and 269 – I suggest to tone down the conclusion. Something like – our analysis does not support the possibility that…
- We have toned down the conclusions as suggested. Please refer to lines 274-275. “Thus, in silico analysis does not support the possibility that miRs or genome DNA methylation are significant regulators of UBQLN4 mRNA levels in a pan-cancer analysis.”
I identified some typos (L stands for line):
- L48 a ubiquitin (however 'an ubiquitin' is also accepted yet less commonly used, so up to the authors. This appears many times in the text).
- We have unified the terminology to “a ubiquitin”.
- L60 – the authors use ubiquitylated, in other parts they use ubiquitinated (e.g. L67). Both are OK, they should choose one of those and be consistent.
- We have unified the terminology to “ubiquitylated”.
- L198 from 48 patients.
- We apologize for the mistake. This has been corrected.
- L278 the abbreviation UCS is not explained anywhere.
- UCS abbreviation was defined in line 519.
- L383 should be: highest sensitivity to cisplatin treatment.
- We apologize for the mistake. We modified the sentence. Please refer to lines 394-396. “the ovarian cancer cell line OVK18, which showed the highest UBQLN4 mRNA levels in CCLE had the highest sensitivity to Olaparib, despite having the lowest sensitivity to cisplatin treatment.”
- L413 sentence not clear to me.
- The sentence was re-written. Please refer to lines 431-432. “We previously demonstrated that in ESCC FFPE tissues from endoscopic core biopsies with high UBQLN4 mRNA expression could predict a worse response to NAC [8].”

Reviewer 2 Report
Using various bioinformatic and statistical analysis, the manuscript presents a compelling case that Ubiquitin-4 overexpression is a prognostic marker for drug resistant cancers. IHC analysis validates that Ubiquitin-4 is highly present in tumor samples compared to normal tissue. The study is overall nice validation of a previous paper from Shiloh lab demonstrating that Ubiquitin-4 overexpression promotes drug resistance.
Several comments
· Do authors have any information that the overexpressed Ubiquitin-4 bears any type of mutations? Or are they wild type with normal activities?
· The paper would be more interesting if more functional assays are added to back up the significance of the UBQLN4 overexpression, which is mostly correlative. For instance, does knocking down UBQLN4 selectively sensitizes the overexpressed cells, compared to non-overexpressed cells? (or, does forced overexpression of UBQLN4 cause drug resistance in otherwise drug-sensitive cancer cells?)
· In the abstract it says “the regulatory mechanism of UBQLN4 overexpression remains unknown”, but the manuscript is not about regulatory mechanism, rather it is more about expression/prediction analysis.
Author Response
Reviewer 2
Comments and Suggestions for Authors
- Do authors have any information that the overexpressed Ubiquitin-4 bears any type of mutations? Or are they wild type with normal activities?
- The reviewer’s comment is important. We would like to acknowledge that the frequency of mutations in UBQLN4 gene are rare. Only few mutations have been described, but there not hotspots in the UBQLN4 Please refer to Figure S3A, where we showed the frequency of mutations is 0.48% (only 194 functional SNVs of the UBQLN4 gene in 40,549 patients for 40 cancer types) and 0.79% (86 functional SNVs of UBQLN4 in 10,953 patients in 32 cancer types) using the Catalogue of Somatic Mutations In Cancer (COSMIC) and TCGA datasets, respectively.
- Nonetheless, in our previous paper we had reported two cases with UBQLN4 mutations that induced a genomic syndrome that resembles genome instability. Please refer to our previous study for more information (Jachimowicz et al. Cell 2019).
- The paper would be more interesting if more functional assays are added to back up the significance of the UBQLN4 overexpression, which is mostly correlative. For instance, does knocking down UBQLN4 selectively sensitizes the overexpressed cells, compared to non-overexpressed cells? (or, does forced overexpression of UBQLN4 cause drug resistance in otherwise drug-sensitive cancer cells?).
- The reviewer is correct. We totally understand the limitations of the in-silico However, the consistency in the observations had made us strongly believe that UBQLN4 is a significant factor associated with prognosis and treatment resistance in different solid tumors. We had previously performed functional studies in our three published studies in Esophageal Squamous Cell Carcinoma (Murakami et al, Mol Oncol, 2021, ref #8), as well as Breast Cancer (Shoji et al, Clin Trans Med, 2022, ref #9), Melanoma and Neuroblastoma (Jachimowicz et al, Cell, 2019, ref #7). In these studies, we have shown the effect of both UBQLN4-OV and UBLQN4 knockout/knockdown on DNA-damage drugs sensitivity. We had also demonstrated the clinical relevance of UBQLN4 upregulation in neoadjuvant treatment responses in ESCC patients. Therefore, our previous recent publications support the functional aspects of UBQLN4 in specific cancer types.
- In the abstract it says “the regulatory mechanism of UBQLN4 overexpression remains unknown”, but the manuscript is not about regulatory mechanism, rather it is more about expression/prediction analysis.
- We have addressed the reviewer’s comment and modified this statement as it is confusing. When we mentioned regulatory mechanisms, we referred to study the mechanism at the DNA level, transcriptionally and post-transcriptional levels that control the upregulation of UBQLN4 mRNA and protein levels. Nevertheless, we have modified this throughout the manuscript. Please refer to lines 20-21, 76-77, 477-478.
Reviewer 3 Report
Genomic amplification of UBQLN4 is a prognostic and treat-ment resistance factor
Yuta Kobayashi , Matias A. Bustos , Yoshiaki Shoji , Ron D. Jachimowicz , Yosef Shiloh , Dave S. B. Hoon
The article summarizes the knowledge about UBQLN4 deduced from the cancer patient’s datasets and provides the empirical data to reason about cancerogenesis. Suggestions to improve the the paper:
- To ensure that the amplification of the gene is the main mechanism of UBQLN4 upregulation in the tumor cells, it is necessary to compare the number of copies of UBQLN4 gene in the tumor cells with the the copies number of the gene in the corresponding normal tissues or in the adjacent normal tissues.
- The authors should mention what was the exact method used to identify the up-regulation mechanism of UBQLN4
Minor:
- Meaning of the CNA abbreviation (copy number amplification) must be explained when first mentioned on the line 113;
- In the text of the article, there are different spellings in-silico analysis and in silico analysis. Style better be unified;
- The abbreviation HCC (line 255) is not explained. Is HCC HepatoCellular Carcinoma? Lines 184-185 give other abbreviations for the similar cancerous tissue.
- The authors should clarify the meaning of the Pearson coefficient in relation to the correlation between variables: no correlation/negative correlation/positive correlation (+1 or -1 depending on whether the relationship is positive or negative, respectively).
- Based on the TCGA data analysis, the authors conclude that patients with high UBQLN4 mRNA levels had significantly lower overall survival expectation than patients with the low UBQLN4 mRNA levels in various malignancies. There is no clarification whether the survival was assessed in patients with or without anticancer therapy? (lines 208-213)
- The results of Figure 4A are difficult to understand without further explanation. It is necessary to provide additional captions and axis legend along with explanations for Figure 4A.
Author Response
Reviewer 3
Comments and Suggestions for Authors
The article summarizes the knowledge about UBQLN4 deduced from the cancer patient’s datasets and provides the empirical data to reason about cancerogenesis. Suggestions to improve the paper:
- To ensure that the amplification of the gene is the main mechanism of UBQLN4 upregulation in the tumor cells, it is necessary to compare the number of copies of UBQLN4 gene in the tumor cells with the copies number of the gene in the corresponding normal tissues or in the adjacent normal tissues.
- Since the focal-level copy number variation values for adjacent normal tissues in TCGA are not publicly available, we could not apply the same approach for the analysis of UBQLN4 copy number in adjacent normal tissues. Then, we obtained the data of somatic copy number alterations in 1,708 adjacent normal tissues from 27 cancer types in TCGA from a previous study by Y. A. Jakubek, et al. in Nat Biotech. 2020. Using this data, we examined the proportion of copy number alterations in adjacent normal tissues.
- In addressing adjacent normal tissue analysis, one has to be cautious whether they are pre-malignant and not truly normal thereby immediate adjacent tissue analysis can be misleading in gene expression. Nonetheless, the overall CNVs in adjacent normal tissues were observed in 4.6% of patients (78 of 1708). As for the chromosome 1q region, 1.1% patients (18 of 1708) had CNVs in 1q. CNVs as gain in 1q in adjacent normal tissues were detected only in 9 patients (0.059%). These results indicated that the copy number in adjacent normal tissues is not amplified and thus the mRNA levels of UBLQN4 are not upregulated compared to tumor tissues. We added the new data in Figure S6 and added a new reference in the manuscript (#25), as well as clarification sentences in lines 121-123 in Materials and Methods section, lines 279-281, 292-295 in Results section and line 487 in Supplemental Materials section.
- The authors should mention what was the exact method used to identify the up-regulation mechanism of UBQLN4.
- To our understanding the reviewer’s question, the empirical data suggested that the most common mechanism of UBQLN4 upregulation is copy number variation. We not only showed that the UBQLN4 CNAs are frequently observed across cancer types, but also analyzed all CNAs of the whole genome region of each cancer type to identify significantly amplified regions using PART method and showed UBQLN4 is contained in one of those regions. The PART method was published in 2012 (Niida et al, Bioinformatics, 2012, ref#26) and still used in recent studies (Saiki et al, Nat Med, 2021, ref#27). Therefore, the answer would be that copy number variation correlates with enhanced UBQLN4 mRNA levels. This is occurring because we have shown that at DNA level the promoter region is unmethylated, allowing the UBQLN4 gene to be transcribed. We did not observe any predicted miRNA that consistently targeted UBQLN4 mRNA, thus allowing for the UBQLN4 mRNA to be transcribed to protein. Finally, we had performed IHC to demonstrated that the protein levels are consistently upregulated in majority of solid tumors.
Minor:
- Meaning of the CNA abbreviation (copy number amplification) must be explained when first mentioned on the line 113.
- We apologize for this and has been corrected in line 115.
- In the text of the article, there are different spellings in-silico analysis and in silico analysis. Style better be unified.
- We apologize for the mistake. This has been unified to in silico.
- The abbreviation HCC (line 255) is not explained. Is HCC HepatoCellular Carcinoma? Lines 184-185 give other abbreviations for the similar cancerous tissue.
- The HCC abbreviation has now been properly defined in line 261. A list of all abbreviations can be found in lines 511-512.
- The authors should clarify the meaning of the Pearson coefficient in relation to the correlation between variables: no correlation/negative correlation/positive correlation (+1 or -1 depending on whether the relationship is positive or negative, respectively).
- We have clarified the meaning of Pearson coefficient in lines 268, 301, and 385.
- Based on the TCGA data analysis, the authors conclude that patients with high UBQLN4 mRNA levels had significantly lower overall survival expectation than patients with the low UBQLN4 mRNA levels in various malignancies. There is no clarification whether the survival was assessed in patients with or without anticancer therapy? (lines 208-213).
- One of the criteria for tissue inclusion before those being accepted by the TCGA consortium was that tissues samples should not have had prior therapies. Therefore, the TCGA tissue samples had not received therapies immediately prior to surgical tissue removal.
- The results of Figure 4A are difficult to understand without further explanation. It is necessary to provide additional captions and axis legend along with explanations for Figure 4A.
- We apologize for the lack of proper explanation. We have re-written the Figure legend and add further explanations for Figure 4A. Please refer to page #12, lines 402-406.
